# Synbiotics and Their Antioxidant Properties, Mechanisms, and Benefits on Human and Animal Health: A Narrative Review

**DOI:** 10.3390/biom12101443

**Published:** 2022-10-09

**Authors:** Majid Mounir, Amal Ibijbijen, Kawtar Farih, Holy N. Rabetafika, Hary L. Razafindralambo

**Affiliations:** 1Department of Food Science and Nutrition, Hassan II Institute of Agronomy and Veterinary Medicine, P.O. Box 6202, Rabat 10112, Morocco; 2ProBioLab, Campus Universitaire de la Faculté de Gembloux Agro-Bio Tech/Université de Liège, B-5030 Gembloux, Belgium; 3BioEcoAgro Joint Research Unit, TERRA Teaching and Research Centre, Microbial Processes and Interactions, Gembloux Agro-Bio Tech/Université de Liège, B-5030 Gembloux, Belgium

**Keywords:** probiotics, prebiotics, synbiotics, antioxidants, metabolites, human health, animal health

## Abstract

Antioxidants are often associated with a variety of anti-aging compounds that can ensure human and animal health longevity. Foods and diet supplements from animals and plants are the common exogenous sources of antioxidants. However, microbial-based products, including probiotics and their derivatives, have been recognized for their antioxidant properties through numerous studies and clinical trials. While the number of publications on probiotic antioxidant capacities and action mechanisms is expanding, that of synbiotics combining probiotics with prebiotics is still emerging. Here, the antioxidant metabolites and properties of synbiotics, their modes of action, and their different effects on human and animal health are reviewed and discussed. Synbiotics can generate almost unlimited possibilities of antioxidant compounds, which may have superior performance compared to those of their components through additive or complementary effects, and especially by synergistic actions. Either combined with antioxidant prebiotics or not, probiotics can convert these substrates to generate antioxidant compounds with superior activities. Such synbiotic-based new routes for supplying natural antioxidants appear relevant and promising in human and animal health prevention and treatment. A better understanding of various component interactions within synbiotics is key to generating a higher quality, quantity, and bioavailability of antioxidants from these biotic sources.

## 1. Introduction

Antioxidants can be a variety of compounds, which are able to neutralize, either directly or indirectly, oxidative agents. These are mainly represented by free radicals and reactive oxygen/nitrogen species (ROS/RNS) [1]. ROS and RNS are responsible for oxidative stress that leads to fast cell aging in humans [2,3,4] and animals [5,6]. Oxidative stress occurs when there is an out of balance between the formation and neutralization of ROS and RNS. To achieve equilibrium, the human (and animal) body reacts with antioxidants from endogenous (metabolic antioxidants) and/or exogenous (nutrient antioxidants) sources. 

Antioxidant properties and activities are assumed to prevent the harmful effects of ROS/RNS, and therefore treat oxidative stress-related diseases. By increasing the body’s antioxidant defenses through consumption of antioxidant-rich food or dietary supplements, many chronic diseases, as well as disease progression, can be prevented and slowed, respectively. Therefore, antioxidants are often associated with anti-aging compounds, which are able to contribute to increasing the longevity of animals and humans [7]. Nevertheless, antioxidants can also become pro-oxidants; that is, they are able to induce oxidative stress by forming reactive species, or by inhibiting antioxidant systems [8]. 

Antioxidants may be classified according to many criteria, depending on their action mechanism, origin, chemical structure, and physicochemical properties. Two main groups are easily distinguished according to their role and function: (1) chain-breaking or primary antioxidants, and (2) preventing or secondary antioxidants [9]. The former group is able to react with radicals and convert them into more stable compounds, therefore neutralizing therefore the oxidation chain reactions initiated by free radicals. The latter is known to decelerate the autoxidation degree by changing free radicals to more stable species. This mechanism involves compounds that bind metal ions, scavenge oxygen, decompose hydroperoxide to non-radical species, absorb UV radiation, or deactivate singlet oxygen. Secondary antioxidants need a second minor component to be active. A third group includes tertiary antioxidants that repair the oxidized molecules through sources such as dietary or consecutive antioxidants [10]. On the other hand, antioxidants are classified based on their chemical nature and structure in two categories: non-enzymatic or enzymatic compounds [1], as illustrated in Table 1. Foods, phytochemicals, and dietetic supplements are the most natural external sources of antioxidants, and their powers and activities may vary from one substrate to another [11]. Antioxidant capacities are often determined and compared among different sources by means of various methodologies and scientific instruments through both qualitative and quantitative approaches [12].

While fruits and vegetables are the most popular sources of natural antioxidants, processed foods including beverages and functional foods also contain antioxidant compounds, which may be different from the native ones issued from raw materials [13,14]. The second popular antioxidant sources are dietetic supplements in which the antioxidant compounds are often of a high purity degree and concentration (e.g., vitamins, omega 3 fatty acids) [15]. In addition, other antioxidant sources come from probiotics, which are found especially in fermented foods, as well as in dietetic supplements [16]. Probiotics are defined as live microorganisms that, when administered in an adequate amount, confer health benefits to the host [17]. Lactic acid (e.g., Lactobacilli) and soil-based bacteria (e.g., Bacilli), as well as yeasts (e.g., Saccharomyces) are among the most common microbial probiotics [18]. 

For a few years now, original research and review papers on the probiotic antioxidant properties and action mechanisms have considerably increased [19,20,21,22]. Lactic acid bacteria (LAB) have been shown to exhibit antioxidant capacity, mainly by scavenging free radicals, chelating prooxidative ions, regulating relevant enzymes, or modulating gut microbiota [23]. 

When probiotics are combined with prebiotics into formulations, the resulting functional products constitute synbiotics. Even though the synbiotic concept was first described 25 years ago, the panel of International Scientific Association for Probiotics and Prebiotics (ISAPP) recently updated the synbiotic definition as “a mixture comprising live microorganisms and substrate(s) selectively utilized by host microorganisms that confers a health benefit on the host” [24]. Such a preparation can be designed in complementarity to target the host microorganisms, or in synergism for which the prebiotic is selectively utilized by the co-administrated probiotics to achieve one or more health benefits. The term synbiotic is often confused with symbiotic, which refers to an ecological relationship in a natural ecosystem with two organisms (the symbiont and the host) in symbiosis. Prebiotics are mainly carbohydrate-based compounds such as galacto-oligosaccharides (GOS), fructo-oligosaccharides (FOS), trans-galacto-oligosaccharides (TOS), inulin and fructans, which can improve the viability of probiotics [18]. Non-carbohydrate-based compounds such as polyphenols and omega-3 long fatty acids are also considered as prebiotics according to the standard definitions [25]. In fact, any compounds selectively used by host microbiota and conferring health benefit(s) are considered as prebiotics. 

While the antioxidant properties of probiotics have widely been reported [16,23,26], only a limited number of scientific publications is available on those of synbiotics. Considering the multi-component and mixture aspects (living and non-living materials) of synbiotics, their action mechanisms related to antioxidant activities are much more complex. In fact, it is important in the case of synbiotics with antioxidant properties to distinguish those from prebiotics, probiotics and their metabolites, or those from bio-converted prebiotic compounds. Two main types and mechanisms may be involved: (i) complementary synbiotics for which prebiotics and probiotics act independently with the additive effect as antioxidants at the host [27]; (ii) synergistic synbiotics where prebiotics are antioxidants or not, while supporting and enhancing the probiotics antioxidant performance for generating higher properties than each component (Figure 1). For instance, non-antioxidant oligosaccharide-based prebiotics, when associated with probiotics, may enhance the antioxidant properties of the mixtures [28]. When prebiotics, e.g., exopolysaccharides (EPS), possess antioxidant activities, these bio-compounds can enhance probiotics performance [29,30]. Another case occurs when antioxidant prebiotics serve as probiotic substrates for producing more powerful antioxidant compounds in the formulated synbiotics. It is, for instance, the case of polyphenols bio-converted by Lactobacilli probiotic strains into compounds with superior antioxidant activities such as protocatechuic acid and catechin [31].

The goal of this review paper is double: (1) reviewing synbiotic antioxidant properties and action mechanisms, which are less developed and more complex, and (2) illustrating their benefits on human and animal health through their antioxidant activities. 

## 2. Antioxidant Properties of Synbiotics

While probiotics have long been acknowledged as beneficial to human health, particularly thanks to their antioxidant properties, research into the role of synbiotic antioxidants is still in its early stages. In fact, the effects of probiotics or prebiotics alone, and especially the interactions of both within synbiotic preparations, are involved in the antioxidant action mechanisms. Each component plays a vital role in neutralizing free radicals. Some probiotics such as *Clostridium butyricum* MIYAIRI 588, *Lactiplantibacillus plantarum* CAI6, and *Lacticaseibacillus rhamnosus* GG have been shown to successfully coordinate redox homeostasis in the host cell, resulting in increased overall antioxidant capacity [16,26,32]. It is also stated that probiotics can influence the redox status of the host by their capacity to: (i) chelate metal ions; (ii) activate the host’s antioxidant system in addition to having its antioxidant enzyme system; (iii) create metabolites with antioxidant activity, such as GSH and butyrate; (iv) mediate antioxidant signaling pathways; (v) regulate enzymes that produce reactive oxygen species; and (vi) regulate the intestinal microbiota [33]. Likewise, the antioxidant properties of prebiotics have been studied and demonstrated, for instance, on goat milk fermented by *L. plantarum* L60 [34]. According to these findings, a sufficient amount of prebiotics, e.g., inulin and FOS, can stimulate goat milk fermentation while increasing the antioxidant activity of fermented goat milk. Furthermore, dietary fiber (DF) and polyphenols are also able to enhance gut flora by assuming prebiotic activities [35]. These compounds are chemically and biologically active plant secondary metabolites with several health benefits. These include the fight against oxidative stress-related issues such as cancers, as well as cardiovascular, inflammatory, and neurological diseases. In both chemical and nutritional investigations, DF compounds and polyphenols were traditionally treated as two distinct sets of food constituent. However, there is sufficient scientific evidence that DF transports a considerable number of phytochemicals associated to the complex dietary matrix, primarily polyphenols [36]. 

### 2.1. Probiotic Components

Probiotics are one of the natural sources of both enzymatic and non-enzymatic antioxidants. These come from intact probiotics cells [37], cell-free and intracellular extracts [20,37], intracellular and extracellular metabolites [38], or cell wall components such as exopolysaccharides (EPS) and proteins [39]. When antioxidants come from probiotic dead cells and fragments, the concept of postbiotics is to be considered instead of probiotics antioxidants. This topic is not treated in this review paper. LAB can release a large panel of metabolites with antioxidant activity through lactic acid fermentation that depends on strains, growth medium components, and enzymatic activity [40]. In particular, LAB are frequently used to produce antioxidant peptides from different protein sources, including plants, animals, marine sources, and industrial by-products [14]. Table 2 lists some identified antioxidant compounds produced by probiotics.

### 2.2. Prebiotic Components

Most prebiotics are carbohydrate compounds, mainly oligosaccharides (e.g., FOS, GOS, POS, XOS, inulin), polysaccharides (e.g., β-glucan, guar gum, pectins), and disaccharides (e.g., lactulose). Other non-carbohydrate compounds such as polyphenols, polyunsaturated fatty acids, and minerals also confer prebiotic activities [56]. Prebiotics such as oligosaccharides occur naturally in dietary food products, e.g., banana, asparagus, barley, chicory, spinach, berries, onion, mushrooms, and so on. There are also new emerging sources of polysaccharides prebiotics such as seaweeds and microalgae [57,58]. EPS from microorganisms namely *L. plantarum* exhibit prebiotic properties, which could be useful for some probiotics [59]. According to ISAAP definition, the main health benefits of prebiotics result from their selective utilization by host microorganisms to release several metabolites such as short chain fatty acids (SCFAs) that influenced host physiology. Table 3 lists some examples of prebiotics developing antioxidant properties.

### 2.3. Synbiotic Components

Synbiotics, as mixtures of live microorganisms and substrates selectively utilized by the host microorganisms, can act in synergy or complementary for multiple functions, including antioxidant activities, to confer a health benefit on the host [23]. Some examples of probiotics and prebiotics associated in common synbiotics are displayed in Table 4.

## 3. Antioxidant Action Mechanisms of Synbiotics

The mechanism underlying the antioxidant capacities of synbiotics has been linked to their ability in activating and translocating nuclear factors. These induce the expression of the antioxidant defence enzymatic system, produce antioxidant key molecules, and detoxify the production of singlet oxygen and free radicals [69,70,71]. Recent research on synbiotic dairy products has also revealed that they contained a variety of key vitamins and regulators. These include water-soluble vitamins, antioxidants, and GSH, an important tripeptide involved in the direct chemical neutralization of singlet oxygen, hydroxyl radicals, and superoxide radicals [69,72]. Another study on diabetic patients recently showed that synbiotics supplementation lowered malondialdehyde (MDA) levels, a lipid peroxidation marker [73], and increased (i) GSH levels, (ii) nitric oxide (NO), as a key intra- and intercellular regulating molecule with a wide range of physiological effects [74], and (iii) total antioxidant capacity (TAC), as an indicator of the amount of scavenged free radicals [71,75].

### 3.1. Probiotics’ Action Mechanisms

Increasing attention has been paid to probiotics’ antioxidant performance through numerous recent in vitro and in vivo studies. The probiotics’ antioxidant properties of intact cells, cell-free and intracellular extracts, intracellular and extracellular metabolites, and cell wall components have all been extensively studied [19,20,21,22,76], as illustrated in Table 5. A list of principal methods for evaluating probiotics’ antioxidant activities are summarized in Table 6. The in vitro methods use the capacity of probiotics to scavenge free radicals (DPPH and ABTS scavenging assay), reduce ferric ions using the ferric reducing power assay (FRAP), inhibit lipid peroxidation (β-carotene bleaching assay), and chelate metals; conversely, most in vivo methods involve enzymatic assays. For instance, the lipid peroxidation inhibition assay using thiobarbituric acid reactive substances (TBARs) and DNA damage evaluation with luminescent biosensors is one of the most commonly used techniques for assessing oxidative cellular damages. In this case, the lipid peroxides produced during the oxidation of phospholipids and polyunsaturated fatty acids (PUFAs) are degraded into MDA and 4-hydroxy-2-noneal (4-HNE), which reflect the degree of lipid peroxidation in the body.

Several modes of action, including scavenging free radicals, increasing antioxidant enzymes levels, chelating metal ions, enhancing host antioxidant metabolites (vitamin B12, GSH, folates, etc.), regulating and mediating of host antioxidant signalling pathway or modulating the microbiota, have been proposed [23]. Most antioxidant properties result from the multiple antioxidant abilities. Nevertheless, two antioxidant action mechanisms of microbial probiotics’ enzymatic and non-enzymatic activities can be distinguished for directly inactivating reactive species. These are achieved through a rapid and sensitive oxidative stress response by increasing the activity of endogenous antioxidase enzymes, excreting metabolites (e.g., EPS, vitamins B12, GSH, folates, compounds with radical scavenging ability, etc.), and chelating prooxidant (e.g., ferrous and copper ions). These metal ions are involved in hydroxyl radical formation by decomposing hydrogen peroxide through Fenton catalysts [22]. Probiotics have also been reported to indirectly control the oxidative stress of the cell host by enhancing antioxidase activity [81], reducing ROS producing enzymes, and regulating the antioxidant signalling pathway [22,79]. As an illustrative example of action mechanisms, SODs are LAB’s important multimeric antioxidant metalloenzymes for which MnSOD is more predominant than FeSOD, CuSOD, or ZnSOD [23]. These enzymes catalyse the transition of O_2_^−^ into H_2_O_2_. *L. fermentum* and *L. paracasei* strains are among LAB exhibiting high SOD activities in vitro and in vivo [98,99]. Another enzyme-based probiotic antioxidant is the heme-dependent CAT that catalyses the decomposition of H_2_O_2_ to H_2_O and O_2_. Although Lactobacilli are CAT-negative probiotics, due to their inability to synthesize heme, CAT activities are stimulated by the heme autolysate of *B. subtilis* in co-culture with Lactobacilli [100]. GPx produced by *L. plantarum* under optimal conditions [101] can also reduce oxidised glutathione, which is responsible for DNA breakage, protein denaturation, and lipid peroxidation. Probiotics can also stimulate the host’s antioxidant system of the host by increasing the efficiency of antioxidase activities, regulating ROS-producing enzymes such as NADPH oxidase, or regulating antioxidant signalling pathways [26,39,91]. *L. plantarum* Y44 exerted antioxidative effects by scavenging oxygen free radicals and activating the nuclear factor-erythroid 2-related factor-2 (Nrf2) signalling pathway in Caco-2 cells, thus protecting against damage caused by 2,2′-azobis(2-methylpropionamidine) dihydrochloride (ABAP) [91]. *L. helveticus* and *L. plantarum* induced changes in renal protein expression level of SOD1, SOD2, and CAT in a rat model, leading to an improvement in specific metabolic parameters and renal antioxidative enzymes in a fructose-induced metabolic disorder [102]. *L. plantarum* Y44 may alleviate oxidative stress by modulating the gut microbiota composition [103]. This strain induced change in microbiota composition, glycerophospholipid levels, and oxidative stress-related indicators. Probiotics can also help the host’s antioxidant system defence by producing and releasing antioxidant metabolites [41,47].

### 3.2. Prebiotics’ Action Mechanisms

Antioxidant activities of some prebiotics have been reported in the literature [59,60,61,62]. Plant-based XOS and POS have the ability to scavenge DPPH and ABTS radicals. Inulin from Jerusalem artichoke root had low DPPH radical scavenging activity, ABTS radical scavenging activity, and ferric reducing power, but significantly improved the antioxidant status of laying hens with a prebiotic supplemented diet, i.e., caused an increase of the enzyme antioxidant activities of SOD, CAT, and GSH-Px. Glucan-based EPS produced by *L. plantarum* have free radical scavenging activities. These activities are attributed to the presence of a hydroxyl group and other functional groups capable of donating electrons to reduce the radicals to a more stable form, or to react with the free radicals to terminate the radical chain reaction. Oligosaccharides have the ability to scavenge different radicals, such as DPPH and ABTS radicals. The hydroxyl groups in positions C-2 and C-6 in oligosaccharides are involved in H-atom transfer reactions with these radicals [104]. NAOS obtained by enzymatic degradation from red algae polysaccharides demonstrated antioxidant activities depending on the degree of polymerisation [63].

### 3.3. Synbiotics’ Action Mechanisms

Taking into account the probiotic and prebiotic combinative effects, either in a complementary or in a synergistic way, there is growing evidence to suggest the antioxidant activities of synbiotics, with a few illustrative examples in human and animal species.

*Lactobacillus* and *Bifidobacterium* strains are thought to be the most significant probiotics involved in synbiotic antioxidant activities [105]. It has been found that synbiotics combining *L. casei* and inulin were efficient substances that protected the human body from the damage caused by free radicals. Synbiotics may improve blood plasma antioxidant capacity and the activity of certain antioxidant enzymes [106]. 

Relevant study has been led on a synbiotic combining the multistrain probiotics VSL#3 (four strains of *Lactobacillus*, three strains of *Bifidobacterium*, and one strain of *Streptococcus*) with the yacon-based product PBY, which contains high concentrated prebiotics FOS and inulin. The probiotics VSL#3 and the synbiotic VSL#3 with PBY had a high ability to trap DPPH radicals in vitro and in vivo, as evidenced by a considerable decrease in hepatic oxidative stress indicators and enhanced catalase activity [33].

Two recent meta-analyses showed that synbiotic supplementation was linked to enhanced antioxidant resistance and antioxidant enzymes. TAC, GSH levels, SOD, and NO levels were all higher with synbiotic (and probiotic) consumption compared to the controls, but MDA levels were lower [107,108]. 

Moreover, a clinical trial on patients with type 2 diabetes has been conducted to study the effect of consumption of synbiotic bread containing *L. sporogenes* and inulin by measuring antioxidant parameters before and after the intervention. Their results indicated that the consumption of synbiotic bread decreased MDA significantly, while TAC, chloramphenicol acetyltransferase (CAT), and GSH remained unchanged [109].

The antioxidant activity of synbiotic supplementation has also been studied in women with migraines, revealing that synbiotic supplementation of 10^9^ CFU of 12 kinds of probiotics with FOS prebiotic for 12 weeks improved oxidative stress, including TAC and NO, and migraine clinical symptoms [110].

Another study concluded that a diet supplemented with organic Zn and a synbiotic combination delayed the lipid oxidation process in piglets throughout the refrigeration phase [111]. It has also been shown that the consumption of synbiotics boosted the antioxidant defense system and reduced lipid peroxidation in the liver of rats by enhancing antioxidant enzymes’ activity and limiting the development of MDA in the liver [112].

## 4. Applications to Human Health

Synbiotic products can be beneficial to the intestinal or extra-intestinal microbial ecosystems of animal and human species through feed additives, foods, non-foods, nutritional supplements, or medications [24].

Beneficial effects of probiotics and synbiotics on oxidative stress-related chronic diseases are generally attributed to their antioxidant properties, alleviating the oxidative stress in organs and DNA damage, reducing inflammation, or enhancing the immune response (Figure 2).

### 4.1. Antioxidative Stress 

Oxidative stress is “an imbalance between oxidants and antioxidants in favour of the former, leading to a disruption of redox signalling and control, and/or molecular damage” [113]. It defines an imbalance condition of the natural defence system prooxidant–antioxidant in cells, i.e., when the total oxidant levels exceeds total antioxidant capacity, resulting in DNA hydroxylation, protein denaturation, lipid peroxidation, and apoptosis [26]. In biological systems, principally endogenous ROS such as superoxide radicals (O_2_^−°^), hydroxyl radicals (°OH), hydrogen peroxide (H_2_O_2_), and lipid peroxide produced during the process of cellular metabolism have been identified to induce these oxidative damages [1]. Other reactive species, namely endogenous RNS such as NO, have been found to produce a deleterious effect on biological systems. Exogenous ROS from exposure to external factors such as pollution, radiation, drugs, bacterial infection, or excessive iron intake are also responsible for oxidative stress [114]. Living cells have a natural defence mechanism to encounter oxidative stress. In order to neutralize the reactive species, biological systems are able to synthesize and release antioxidants such as glutathione and vitamin C, or antioxidant enzymes such as SOD, CAT, and peroxidases [13].

ROS, including superoxide anions, hydroxyl radicals, and hydrogen peroxides, are critical signaling molecules with important roles in many diseases. A variety of chronic and degenerative diseases, as well as the aging process, but also acute pathologies such as neurodegenerative diseases, cancers, cardiovascular diseases, and chronic inflammation, may be attributed to the oxidative stress phenomenon. Both endogenous and exogenous ROS cause oxidative modification of cellular macromolecules (carbohydrates, lipids, proteins, and DNA), leading to lipid peroxidation, protein misfolding and aggregation, DNA damage, and mutations. There are two major mechanisms through which oxidative stress contributes to diseases. The first involves the production of reactive species during oxidative stress, particularly •OH, ONOO^−^, and HOCl^-^ that directly oxidize macromolecules, including membrane lipids, structural proteins, enzymes, and nucleic acids, leading to aberrant cell function and death. The second mechanism of oxidative stress is aberrant redox signalling [115]. The involvement of free radicals in neurodegenerative diseases is largely reported in the literature. Owing to the high consumption of oxygen and enrichment in PUFA, the brain is the most vulnerable part of the body. ROS causes a damaging effect on neurons and accumulates in the brain, resulting in neurodegenerative diseases [116]. The central role of mitochondrial ROS and heart disease is highlighted by a number of genetic models in which the modulation of either mitochondrial ROS production pathways or mitochondrial ROS scavenging systems has a significant impact on cardiac physiology and the development of cardiac diseases [117].

The anti-oxidative stress effects of consumption of a probiotic mix (*B. longum* CECT 7347, *L. casei* CECT 9104, and *L. rhamnosus* CECT 8361) for 6 weeks have been observed in male cyclists under high-intensity and duration physical exercises. The reduction of lipid-related oxidative stress biomarkers, such as serum MDA, serum oxidized low-density lipoprotein (Ox-LDL), and DNA-related oxidative stress biomarkers, such as urinary 8-hydroxy-2′-deoxyguanosine (8-OhdG), is not attributed to the increase in antioxidant enzymes [118].

In Alzheimer’s patients, a continuous dietary supplementation of synbiotic kefir milk had a positive effects on systemic oxidative stress and led to a significative decrease in protein oxidation [119].

### 4.2. Anti-Aging Effects

The free radical theory of ageing (FRTA) states that the organism ages because of free radical-induced cell damage accumulation over time [120]. There is evidence that probiotics and synbiotics are effective in counteracting oxidative stress and DNA damage in cells. *L. plantarum* GKM3 delayed the process of aging, alleviated age-related cognitive impairment, and reduced oxidative stress in mice models [121]. Recent findings suggest that *L. plantarum* JBC5 activated the p38 MAPK pathway and its downstream targets in worms (*Caenorhabditis elegans*) to enhance longevity by improving stress resistance, immunity, and other age-associated pathologies [122]. Other probiotics, such as *B. amyloliquefaciens* B-1895 and *B. subtilis* KATMIRA1933 also induce DNA protective and antioxidant activity [123].

The effects of the synbiotic composed of *L. fermentum* probiotic bacteria and the green tea epigallocatechin gallate (EGCG) on immune rejuvenating effects during aging in aged Swiss albino mice showed evidence of additive effects in the amelioration of oxidative and inflammatory stress-induced cell death. In vivo supplementation of synbiotics significantly enhanced neutrophil oxidative index, CD3+ cell numbers and activation status, Th1/Th2 cytokines in splenic supernatants, as well as liver Nrf-2 expression compared to treatments with *L. fermentum* or EGCG alone [67]. 

### 4.3. Heavy Metal Anti-Toxicity Effects

The detoxification role of probiotics caused by heavy metals has been largely related in the literature to their heavy metal surface binding capacity [124]. Recent findings also highlight their role in heavy metal antitoxicity. For instance, Bifidobacterium sp. MKK4 and its synbiotic rice fermented beverage prevented arsenic toxicity by inducing higher levels of SOD and CAT, and reduced GSH in rat models [125]. Protective actions against mercury toxicity of two synbiotic diets (*B. coagulans* and *L. plantarum* with inulin) in rat models have been shown effective in reducing mercury content in the animal kidney and liver through chelation mechanisms [126].

### 4.4. Prevention and Treatment of Chronic Diseases

Several clinical studies suggest that probiotics and synbiotics may be helpful for preventing and treating various diseases [68]. There is evidence that probiotics/synbiotic supplementation is effective in reducing oxidative stress levels, and thus preventing or ameliorating diabetes, cardiovascular disease, cancer, and other chronic diseases. A meta-analysis on the effects of probiotics/synbiotic supplementation compared to placebo on biomarkers of oxidative stress such as TAC, GSH, MDA, and NO in adults highlighted a significant increase in serum GSH, NO, and TAC, and a significant reduction of MDA levels in the body by probiotics/synbiotic supplementation [127]. Some recent studies on the beneficial effects of antioxidant properties of synbiotics are summarized in Table 7. 

Among examples of LAB probiotics’ effects on diseases, *L. salivarius* AP-32 supplementation in rats with 6-hydroxydopamine (6-OHDA)-induced Parkinson’s disease enhances the host antioxidant enzymes’ activity and SCFA production, inducing protection of dopaminergic neurons, and improvement of motor functions. The supplements also modulate faecal microbiota composition. Some specifically enriched commensal taxa correlate positively with SOD, GPx, and CAT activity, indicating that supplementation also promotes antioxidant activity via an indirect pathway [128]. *L. plantarum* 200655 exhibits radical scavenging activity and lipid peroxidation inhibition activity [96]. An enhancement of immunity was observed on macrophage-like RAW 264.7 cells, which was correlated to a high NO production and high cytokine production of IL-1b and IL-6. Recent similar results have been noticed in novel probiotics such as *Levilactobacillus brevis* KU15147 isolated from radish kimchi. The strains exerted immune-enhancing effects in the stimulation of RAW 264.7 cells, and showed higher cytokine production of inducible NO synthase (iNOS) and tumour necrosis factor-α (TNF-α), in comparison with non-stimulated control cells with LPS [129]. The protective effects of LAB on cisplatin (CP)-induced renal damage have been also observed and attributed to the anti-inflammatory and antioxidant properties of probiotics by decreasing oxidative stress, inflammation, apoptosis, DNA, and histopathological damage in rat kidney tissue [130]. 

Other probiotic genera such as Streptococcus and Bacillus have positive effects on diseases due to their antioxidant properties. *S. thermophilus* YIT 2001 has been shown in a clinical trial to have inhibitory effects on the oxidation of LDL and the development of aortic fatty lesions in an animal model. Such probiotics have the ability to lower the serum levels of MDA-modified LDL, an oxidative modification product of LDL. The intracellular reduced GSH has been associated with the antioxidant activity against LDL oxidation in a hyperlipidaemia hamster model [89]. *B. amyloliquefaciens* ssp. *plantarum* IMV B-7142 and *B. amyloliquefaciens* ssp. *plantarum* IMV B-7143 have hepatoprotective effects against the toxic effects of carbon tetrachloride (CCl4) [88]. 

Concerning the effects of synbiotic supplementation on health, the combination of LAB with fiber (inulin, β-glucan) and oligosaccharides (FOS, XOS) is the most studied. One relevant example is the probiotics mix VSL3 # and its synbiotic association with yacon-based product rich in FOS and inulin, and their protection effects on mucosa from damage caused by chemical carcinogen and reduced intestinal permeability in mice induced to colorectal carcinogenesis. The CAT enzyme activity increases in synbiotic and probiotic groups compared to the control group, while the oxidative stress biomarkers such as MDA and carbonylated protein decreases [33]. One study evaluated the effect of the synbiotic composed of probiotic *B. infantis* and the prebiotic XOS against ulcerative colitis in colitis-induced mice compared to probiotics or prebiotics alone. All treatments significantly inhibited oxidative stress and downregulated the pro-inflammatory cytokines TNF-α and interleukin-1β (IL-1β), and synbiotic treatment significantly upregulated the anti-inflammatory cytokine interleukin-10 (IL-10) in the colon tissues. The synbiotic treatment has been the most efficacious in decreasing the disease activity index and pathological scores against colitis, explained by the additive combination of the direct anti-inflammatory effects of the probiotics and prebiotic components, and their ability to fortify colonic epithelial barrier integrity [131]. The anti-inflammatory and antioxidant effects of the probiotics *L. rhamnosus* GG, the prebiotic oat β-glucan (OAT), and synbiotics (OAT + *L. rhamnosus* GG) against high-fat diets have also been evaluated in mice by examining the fatty acid profiles and oxidized PUFA in the gut–liver–brain axis. The synbiotic composed of *L. rhamnosus* GG and OAT synergistically attenuated the increase in non-enzymatic oxidized products in mice fed with high fat diet, indicating their synbiotic antioxidant property [132]. The original synbiotic association of *L. acidophilus* and cinnamon powder, as well as each component, induced a moderate increase in the level of antioxidant enzymes in patients with type 2 diabetes, the most significant change being observed within the probiotics group [133]. 

## 5. Applications to Animal Health

Probiotics and synbiotics are potential bioagents that can be used to treat any veterinary animal disease, or simply improve their health. The following section discusses the applications of probiotics and synbiotics in instances where scientific evidence supports their use for their antioxidant properties, or to meet other needs for each main group of farming animals.

### 5.1. Poultry

Probiotics and synbiotics are used in feed additives to enhance the effectiveness of nutrients and improve poultry’s performance [137]. Probiotics have the ability to substitute antibiotic growth promoters (AGP), which are commonly utilized by poultry farmers today. They aim to keep broiler chicks healthy and enhance their development potential. AGP in feed has been linked to intestinal bacterial resistance, as well as antibiotic residues, in broiler chicken meat. As a result, practically every country in the world today prohibits the use of AGP. 

In poultry, a range of bacteria and yeast species have been studied and utilized as probiotics. The majority of the research was focused on analyzing the benefits of probiotics in lowering the number of pathogenic bacteria in the gastrointestinal tract (GIT), as well as the effects of probiotics on boosting growth and performance in disease-free chickens. In broiler chickens, adding a single or multistrain of *Lactobacillus* sp. to broiler chicken feed increased their body weight and feed efficiency. Probiotics based on *Bacillus* sp. have also been proven to be beneficial in chicken diets and were found to promote animal growth [138]. A study was conducted to see the effect of a multispecies probiotic-based feed containing *Lactococcus lactis*, *Carnobacterium divergens*, *L. casei*, *L. plantarum,* and *S. cerevisiae* on the reduction of *Campylobacter* spp. infection rates in broiler chickens raised on a commercial farm. The results of this study demonstrated that adding probiotics (Lavipan) to a broiler chicken feed reduced the extent of *Campylobacter* spp. invasion in the birds’ gastrointestinal tract and, as a result, reduced contamination levels in the birds’ environment, contributing to improved hygienic parameters of the analyzed poultry carcasses. Furthermore, probiotics showed promising immunomodulatory capabilities, which might help increase the efficacy of a particular prophylactic program used in a flock of broiler chicks [139]. Another study looked at the impact of screened LAB strains on broiler chicken development, humoral immunity, and IGF-1 gene expression. In comparison to the control group, probiotic diets significantly improved feed conversion ratio, increased body weight, and raised carcass relative weight. The lymphocyte count was also much higher, while serum triglycerides and total cholesterol levels were significantly lower. *Lactobacillus* spp. populations increased substantially, while *Escherichia coli* populations decreased significantly, and the expression of the IGF-1 gene in broiler liver tissue was significantly increased compared to the control group [140]. A study was carried out to examine the competitive exclusion of *Campylobacter jejuni* in poultry gut by three potential probiotic Lactobacilli strains [141]. *L. gallinarum* PL 53 was found to be an effective probiotic, exhibiting competitive exclusion of *C. jejuni* and significantly lowering microbial load in an in vivo trial experiment, as well as maintaining the overall health of the gut microbiota by preventing a variety of potential foodborne pathogens. At the primary production stage, *L. gallinarum* PL 53 inhibited *C. jejuni* colonization. However, a recent study evaluated the effects of two commercial probiotics (Pro-Biotyk and Em-15, EMFarma™) on body weight, feed intake and conversion, carcass characteristics, and microbial contamination in a hen house. The probiotic formulations resulted in an insignificant increase in body weight, feed intake, and feed conversion ratio after 4 weeks of growing the chickens, as well as an insignificant decrease in chicken mortality. Pre-slaughter body weight, carcass weight, dressing percentage, and carcass component composition were not substantially different in probiotic-fed chickens. When compared to control chicken carcasses, experimental chicken carcasses had a smaller proportion of breast muscle, leg muscle, abdominal fat, and neck, as well as a larger percentage of skin with subcutaneous fat, wings, and the remainder of the carcasses [142].

Regarding the mechanisms of action, lipid peroxidation is one of the most common causes of meat quality degradation in chicken, and it can (i) reduce nutritional value, (ii) produce taste and texture issues, and (iii) change the look of the meat [143]. MDA is the major end product of ROS lipid peroxidation, and its accumulation is commonly employed to assess the lipid oxidation rate in poultry meat. Supplementing with a synbiotic reduced MDA accumulation in the thigh muscle and fought against meat oxidation, thus improving meat quality and shelf life [144]. Likewise, after 30 days of storage at 4 °C, the value of TBARs in thigh meat decreased linearly as the synbiotic inclusion concentrations in the meals increased [145]. Furthermore, it was discovered that a synbiotic-supplemented diet reduced MDA levels in broilers. The addition of bee pollen and propolis extracts in feed mixtures, in combination with probiotics added into drinking water for broiler chickens, also reduced oxidative processes in the breast and thigh muscles during 7-days of chilling storage [146,147]. 

Due to the availability of their specific substrate for fermentation, synbiotics can improve the survival of the health-promoting microorganisms in birds’ guts, and may have a positive impact on feed absorption and utilization, daily body weight increase, and meat and egg quality [148]. Synbiotics also provide clinical benefits for chickens; these benefits include inhibiting the proliferation of pathogenic bacteria, maintaining the intestinal barrier, modulating immune function, and fighting diarrhea [149]. A dietary supplementation with synbiotic or synbiotic plus organic acid can be used as a potential tool to improve growth performance and reduce carcass *Salmonella* in broilers [150]. As an alternative to antibiotics, the addition of turmeric and synbiotic combination in the diets positively influenced haemato-biochemical parameters and comparative economics with reduced mortality of the broiler [151]. Another study sought to determine the impact of newly developed synbiotic preparations on chicken performance, and found that synbiotics had a positive impact on chicken performance parameters, as well as an increase in the number of beneficial bacteria and a reduction in the growth of potential pathogens in the gastrointestinal tract. In the excreta of broilers, synbiotics increased the concentrations of lactic acid and short chain fatty acid (SCFA), while decreasing the concentration of branched chain fatty acid (BCFA). These findings revealed that the studied synbiotics had a positive impact on the intestinal microbiota, metabolism, and broiler chicken performance [152].

### 5.2. Pigs 

Maternal probiotics or synbiotic supplementation to sows during gestation and lactation significantly enhanced their systemic and intestinal antioxidant capacity, improved mitochondrial biogenesis, and altered the jejunal and colonic bacteria communities in offspring piglets [153]. Furthermore, a correlation analysis indicated that the abundances of antioxidant enzymes and mitochondrial biogenesis-related indices were strongly linked with jejunal and colonic microbiota abundances. Another recent study investigated the effect of a synbiotic on the oxidative stability of lipid in piglets meat, concluding that the diet supplemented with organic Zn and a synbiotic mixture contributed to the delay of the lipid oxidation process of the shoulder and ham samples during the refrigeration period [111].

In addition to their antioxidant properties, probiotics and synbiotics can be of help in other aspects of pig health. In fact, weaning, as it is now practiced, is one of the most crucial phases for pigs, as it is marked by a decrease in food consumption, which can lead to severe anorexia, increase susceptibility to digestive diseases, development delays, and microbial infections. Alterations in the dietary substrate cause significant changes in the intestine’s functioning. Positive alternatives appear to be *S. cerevisiae* yeasts, their cell walls, or extracted fractions. When employed in piglets’ diet, they can promote growth, activate the immune system, maintain the balance of digestive microflora, and limit bacterial adhesion to intestinal epithelial cells. For swine, yeast or yeast derivatives might be a viable alternative to antibiotic growth boosters [154]. Live yeast (LY, *S. cerevisiae* strain CNCM I-4407, 1010 CFU/g) or *S. cerevisiae* coupled with ZnO (LY-ZnO) could replace antibiotics by increasing pigs’ average daily gain, serum IgA, IgG, SOD, fecal butyric acid, and total volatile fatty acid concentrations, and decreasing feed conversion ratio and diarrhea rate compared to the control group [155]. A similar study was conducted to see how incorporating the yeast *S. cerevisiae* or its cell wall fraction into weanling piglet diets affected growth performance, food utilization, and several morphological and immunological characteristics. Overall, yeast diets resulted in increased weight growth and ultimate body weight, as well as an improved feed:gain ratio. The addition of yeasts or yeast cell walls reduced the frequency of intraepithelial lymphocytes while increasing VFA synthesis and acetate percentage, resulting in improved piglet productivity after weaning [156]. In a recent study, pigs’ longissimus thoracis (LT) was examined by ref. [157]. After a 175-day dietary treatment with *L. reuteri* 1 (LR1) and antibiotics (olaquindox and aureomycin), results showed that LR1 (i) reduced drip loss and shear force, (ii) increased inosinic acid and glutamic acid, which may improve flavor, and (iii) changed muscle fiber properties, all of which improved pork quality when compared to the use of antibiotics. Dietary supplementation with live yeast *S. cerevisiae* to sows and piglets throughout late gestation, suckling, and postweaning periods can help reduce the length and severity of *E. coli*-induced postweaning diarrhea. In yeast-fed weaned piglets, reduced infection-related stress and severity of diarrhea can improve growth performance in the pre-weaning phase. *S. cerevisiae* (strain CNCM I-4407) could be used to prevent and treat postweaning diarrhea. In addition, *S. cerevisiae* can reduce, in porcine intestinal epithelial cells, the inflammatory responses generated by F4+ enterotoxigenic *E. coli* [154]. A recent study found that a low-nutrient-density diet supplemented with a probiotic mixture improved the growth performance, faecal microbial content, and faecal gas emission of weaner pigs [158].

### 5.3. Ruminants 

In contrast to chickens and pigs, studies on probiotic and synbiotic antioxidant capacities in ruminants have received less attention. However, in sheep, goats, and cattle, oral probiotic supplementation has been demonstrated to boost feed intake, daily weight gain, and overall weight gain [159]. In dairy cows, probiotics containing live yeast boosted food intake, improved feed efficiency, average daily gain and total weight, and increased milk yield and quality [159,160]. In a more recent study, the effects of probiotics and prebiotics alone or in combination in the diet of lambs finished under subtropical climate conditions have been tested [161]. These researchers found that supplementing finishing lambs with probiotics and prebiotics in subtropical climates may assist in reducing the unfavorable effects of high ambient heat load on dietary energy utilization. Lambs fed with probiotic and/or prebiotic-based supplements showed higher gain efficiency and a lower ratio of observed-to-expected diet net energy compared to controls, with little influence on carcass features, whole cuts, or visceral mass. Supplemental prebiotics were found to be more effective than probiotics ones under the conditions used in this study, but the combination of the two resulted in a larger response in live weight growth. A similar study conducted on goats to evaluate the effects of *S. cerevisiae*, *C. butyricum* and their combination on rumen fermentation and growth performance of heat-stressed goats showed that supplemental probiotics may be an efficient way to reduce the negative effects of heat stress [162].

### 5.4. Aquaculture

Under stressful situations, fish experience oxidative stress, resulting in the formation of reactive oxygen metabolites and peroxides that cause lipid peroxidation and excessive MDA production [163]. High levels of MDA threaten the functionality of body tissues and cells and pose a risk of DNA damage [164]. A diet of *Pediococcus acidilactici* (PA) and pistachio hulls-derived polysaccharide (PHDP) with PA used as a synbiotic reduced MDA levels in Nile tilapia, thus improving the diet’s antioxidant capacity [165]. Ahmadifar and collaborators also discovered that zebra fish (*Danio rerio*) fed with dietary PA have higher antioxidative ability [166]. 

In addition to their evident antioxidant effects, probiotics and synbiotics can promote animal health as antioxidants through indirect action mechanisms. Owing to their capacity to improve two fundamental critical variables of growth performance and disease resistance, contemporary probiotic bacteria may easily fulfill the demands of sustainable aquaculture development [167]. *Lactobacillus* sp. used as probiotics simultaneously eliminate nitrogen and pathogens from polluted shrimp farms [168]. In fact, nitrogenous compounds provoke concerns in the aquaculture system because they are known to be extremely hazardous, and cause mass mortality [169]. It has been shown that some commercial probiotics (AquaStar^®^, EM^®^, and MicroPan^®^) used as water additives can enhance water quality, fish performance, blood biochemistry, immunity, and up-regulate the expression of growth-related genes in Nile tilapia [170]. Table 8 summarizes the different applications of evident synbiotic antioxidant activities in animal health.

## 6. Conclusions

Synbiotics combine probiotics and prebiotics in mixed preparations. They are substantial natural and exogenous sources of antioxidants through fermented foods, feeds, and diet supplements. They are used for preventing, and even treating animal and human age-related diseases. Probiotic and prebiotic antioxidant activities arise from various metabolites and compounds, including cell components, fragments, and extracts. On the other hand, synbiotics’ antioxidant capacities are the consequences of microbial probiotics, compound prebiotics, or both activities, through complementary or/and synergistic interactions. Their common action mechanism is to directly or indirectly neutralize oxidative agents, causing oxidative stress. In its turn, oxidative stress leads to many diseases, owing to fast aging within animal and human cells. Probiotics develop enzymatic and non-enzymatic antioxidant mechanisms for inactivating reactive species by increasing the activity of endogenous antioxidase enzymes, excreting metabolites such as EPS, vitamins B12, GSH, folates, with radical scavenging ability, or chelating prooxidant metal ions. For synbiotics where live probiotics are combined with prebiotic substrates, antioxidant activities may result from almost unlimited possibilities, owing to the variety of existing microorganisms and substrate sources, but also to the cell factory roles of probiotics. Either combined with antioxidant prebiotics or not, live microorganisms are able to convert substrates to generate antioxidant compounds with superior activities. Based on the literature overview, relative synbiotic-based new routes for supplying natural antioxidants appear relevant and promising in animal and human health prevention and treatment. A better understanding of the interactions between pre- and probiotic components within synbiotics, but also those of such components to the host, is a key factor to generating a higher quality, quantity, and bioavailability of antioxidants from these biotic sources. In this context, the best approaches for developing research in such a field are to continue the antioxidant activity screening of a large number of substrates from plants, animals, and microorganisms in vitro, and especially in vivo. Analytical tools must be used in a complementary way for identifying and measuring new antioxidants, as well as their contents in various materials. Understanding their action mechanisms in a wide range of physicochemical conditions, for instance, through the structure–activity relationship study, appears to be the best route for their rational use in the future. Such investigations naturally require multidisciplinary research approaches, including biology, chemistry, and physics for fundamental aspects and high technology for further industrial perspectives.

## Figures and Tables

**Figure 1 biomolecules-12-01443-f001:**
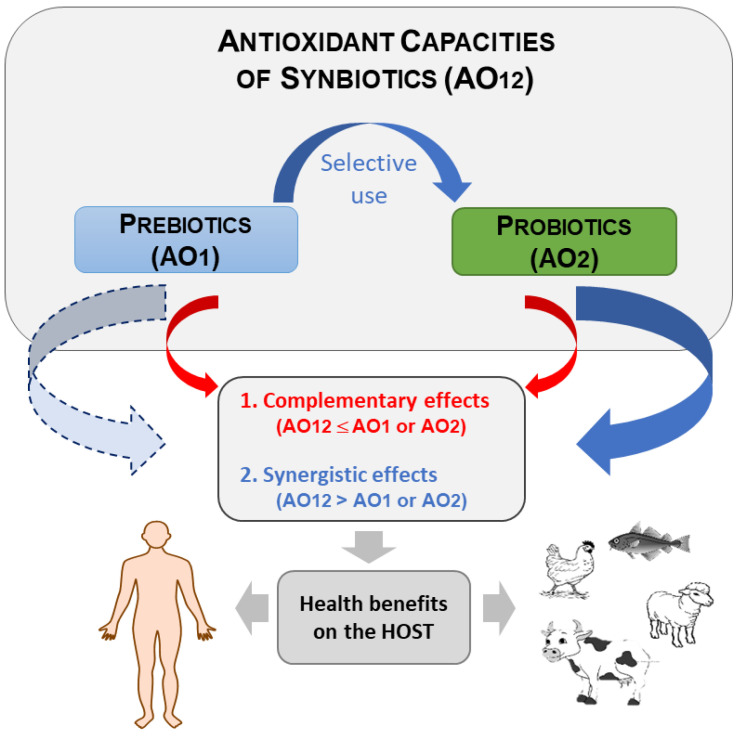
General concept of antioxidant activities (AO) of synbiotics for promoting health benefits on the host (1, 2, and 12 refer to pre-, pro-, and synbiotics).

**Figure 2 biomolecules-12-01443-f002:**
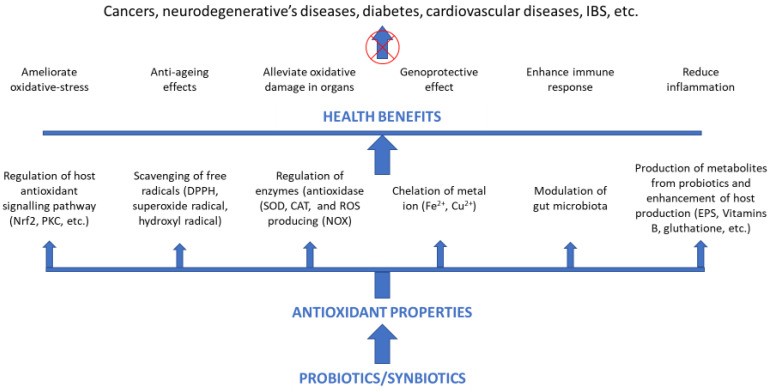
Probiotic and synbiotic antioxidant effects on human health.

**Table 1 biomolecules-12-01443-t001:** Examples of antioxidant categories, symbols, and chemical structures.

Antioxidant Categories	Symbol/Structure
Enzymatic antioxidants	
Superoxide dismutaseCatalaseGlutathione peroxidaseGlutathione reductase	SODCATGPxGRx
Non-enzymatic antioxidants	
*Endogenous (metabolic antioxidants)*
Lipoic acid	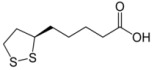
Glutathione (GSH)	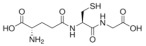
L-arginine	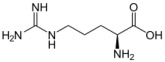
Co-enzyme Q10	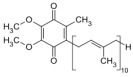
Melatonin	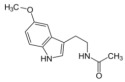
Uric acid	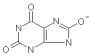
Bilirubin	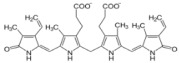
*Exogenous (nutrient antioxidants)*
Vitamin E	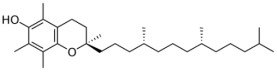
Vitamin C	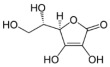
Carotenoids	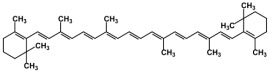
Trace of metals	Se, Mn, Zn
Flavonoids	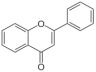 Flavone	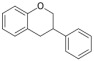 Isoflavone	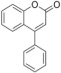 Neoflavonoid
Omega-3 and -6 fatty acids	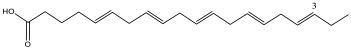 Eicosapentaenoic acid (EPA, C20:5, omega-3)

**Table 2 biomolecules-12-01443-t002:** Examples of identified antioxidant molecules from probiotics.

Antioxidant Molecule	Probiotic Strains	Conditions and Yields	References
Butyrate	*Lactobacillus acidophilus* MG5228	MRS broth 37 °C–overnight 80.70 ± 3.63 µg/g	[41]
Carotenoids C30 carotenoid 4,4′-diaponeurosporene	*Lactiplantibacillus plantarum* subsp *plantarum* KCCP11226	MRS broth 20 °C–24 h 0.74 ± 0.2 A_470_	[42,43]
EPS	*Lactobacillus helveticus* MB2-1	Medium (3g MgSO_4_ + 80 g/L lactose + 20 g/L soya peptone) 37 °C 753 mg/L	[44,45]
	*Streptococcus thermophilus* CS6	Skimmed milk medium	[46]
	*L. plantarum* CNPC003	MRS broth + FOS 37 °C–24 h 568.4 mg/L	[47]
Ferrulic acid	*LimosiLactobacillus fermentum* NCIMB 5221	MRS + ethyl ferrulate 1.33 M 37 °C–24 h 0.168 ± 0.001 mg/L	[48]
Folates	*Enterococcus lactis* BT161	MRS broth 37 °C–overnight 384.22 ± 5.00 ng/mL	[49]
GSH	*Saccharomyces cerevisiae* KU200278 and KU200281	Yeast mold media 25 °C–48 h 5.55 ± 0.52 µg/mg	[21]
	*L. plantarum*	MRS broth as a basal medium + NaCl (5%) + H_2_O_2_ (0.05%) + sodium dodecyl sulphate (0.05%) + amino acids (0.0281%) + urea (0.192%) 40 °C–24h–pH 8 152.61 µM/g	[50]
Hyaluronic acid	*Strep. thermophilus* TISTR 458	Yeast extract 30, K_2_HPO_4_ 2.5, NaCl 2.0 and MgSO_4_•7H_2_O 1.5 g/L, using sugarcane molasses as carbon source 37 ± 2 °C–pH 6.8 213.44 ± 76.79 mg/L	[51]
Levan (EPS)	*Bacillus subtilis*	Yeast extract 2.0g/L, KH_2_PO_4_ 1.0g/L (NH_4_)_2_SO_4_ 3.0; MgSO_4_.7H_2_O 0.06, MnSO_4_ 0.02 and distilled water sucrose 400 g/L 37 °C–16 h 111.6 g/L	[52]
Peptides	*L. helveticus* NK1, *L. rhamnosus* F, *Limosilactobacillus reuteri* LR1	Reconstituted skim milk 37 °C–72 h Not determined (nd)	[53]
	*B. subtilis* MTCC5480	Solid state fermentation; moisture 46% inoculation size, 5.8 × 10^9^ spore/g peptone 5 mg/g and glucose 10.7 mg/g 36 °C–54 days–pH 6.0 369.4 mg/gdp	[54]
Polyphenolic compounds	*S. cerevisiae* var. *boulardii* NCYC 3264	Medium containing 1% (*w*/*v*) yeast extract, 2% (*w*/*v*) peptone, and 2% (*w*/*v*) glucose 30 °C–overnight nd	[20]
Riboflavins (Vitamin B2)	*B. subtilis* subsp. *subtilis* ATCC 6051	Medium (38.10g/L fructose + 0.85 g/L MgSO_4_ + 2.27 g/L K_2_ HPO_4_ + 0.02 g/L FeSO_4_ + 4.37 g/L yeast) 30 °C–72 h 11.73 ± 0.68 g/L	[55]

MRS: De Man, Rogosa and Sharpe, GSH: Glutathione, EPS: exopolysaccharide

**Table 3 biomolecules-12-01443-t003:** Some prebiotics with antioxidant properties.

Class	Prebiotics	Source	Reference
**Carbohydrates**			
Oligosaccharide	POS	Okra	[60]
	XOS	Agricultural wastes (sugar cane straw, coffee husk)	[61]
	Inulin	Jerusalem artichoke root	[62]
	Neoagaro oligosaccharides (NAOS)	Red algae	[63]
Disaccharides	Lactobionic acid	Whey	[64]
Polysaccharides	EPS	Microorganism (*L. plantarum*)	[59]
	Non-starch polysaccharides (arabinoxylan, mannan, arabinogalactan, glucomannan)	Wheat malt beer	[65]
**Non-carbohydrates**			
Polyphenols	Anthocyanins	Purple sweet potato	[66]

**Table 4 biomolecules-12-01443-t004:** Common probiotic and prebiotic components of synbiotics [18,67,68].

Probiotic Genius Bacteria	Prebiotics
*Lactobacillus* *Lactococcus* *Leuconostoc* *Enterococcus* *Streptococcus* *Bifidobacterium* *Saccharomyces* *Bacillus*	Inulin β-glucans Fructooligosaccharides (FOS), galactooligosaccharides (GOS), transgalactooligosaccharides (TOS) Lactulose Polydextose Chicory root inulin-derived (FOS) Wheat bran-derived arabinoxylooligosaccharides (AXOS) Xylooligosaccharides (XOS) Polyphenols

**Table 5 biomolecules-12-01443-t005:** Recent studies on antioxidant properties of probiotics in vitro and in vivo.

Probiotic Strains	In Vitro	In Vivo	Reference
* **Lactobacillus** * **spp.**			
*L. acidophilus*		Stimulation of SOD and catalase activities in carp	[77]
*Lacticaseibacillus casei* NA-2	EPS from probiotics showed antioxidant activities by scavenging hydroxyl radicals (42% at 1.2 mg/mL), superoxide radicals (76% at 100 µg/mL), and 2,2-diphenyl-1-picrylhydrazyl (DPPH) (80% at 10 mg/mL) of EPS		[78]
*L. fermentum* JX306		Improve the activity of GPx, and TAC in the serum, kidney, and liver of D-galactose-induced aging mice model Upregulate the transcriptional level of the antioxidant-related enzyme genes (peroxiredoxin1 (Prdx1), GRx, GPx1, and thioredoxin reductase (TR3) encoding genes)	[79]
*L. helveticus* KLDS1.8701	Strong scavenging properties on DPPH radical, superoxide radical, hydroxyl radical, and chelating activity on ferrous ions	Attenuation of oxidative status (decrease of organic index, liver injury and liver oxidative stress), mitigate hepatic oxidative stress by manipulating the gut microbiota composition in D-galactose-induced mice	[80]
*Lacticaseibacillus paracasei* M11-4	High radical scavenging activities, lipid peroxidation inhibition, and reducing power, antioxidant enzyme activities in the cell-free extract and bacterial suspension	Alleviate D-galactose-induced oxidative damage in the liver and serum of D-galactose-induced rats; prevent D-galactose-induced changes to intestinal microbiota in rats	[22]
*L. plantarum* NJAU-01		High TAC; increase of antioxidant enzymatic activities of SOD, GPx, and CAT in serum, heart, and liver of mice	[81]
*L. rhamnosus* ARJD	Significant nitric oxide (NO) scavenging, hydroxyl radical scavenging activity, DPPH scavenging activities, and reducing power activity	Gastrointestinal stress tolerance abilities with long resident abilities in the host (rat) gastrointestinal tract	[82]
*L. reuteri* MG505	High DPPH free radical scavenging and 2,2′-azinobis 3-ethylbenzothiazoline-6-sulfonate (ABTS) radical scavenging		[83]
* **Bifidobacterium** * **spp.**			
*B. adolescentis* MC-42		Lower oxidative process in hypoxified rat brain tissues	[84]
*B. animalis* subsp. *lactis* MG741	High DPPH free radical scavenging and ABTS radical scavenging		[83]
*B. breve* MG729	High DPPH free radical scavenging and ABTS radical scavenging		[83]
*B. longum* LTBL16	DPPH scavenging ability and oxygen resistance		[85]
* **Bacillus** * **spp.**			
*B. coagulans* MTCC5856	DPPH radical scavenging activity; intracellular ROS scavenging activity		[86]
*B. subtilis* AF17	DPPH radical-scavenging capacity; reducing power; strong total antioxidant activity		[87]
*B.amyloliquefaciens* ssp. *plantarum* IMV B-7143	Stabilisation of the DPPH radical to its neutral form	Protection of stress-damaged rat hepatocytes	[88]
* **Saccharomyces** * **spp.**			
*S. cerevisiae* KU200278 and KU200281		Protection against DNA damage	[21]
*S. cerevisiae* var *boulardii*	DPPH radical scavenging activity		[20]
* **Streptococcus** * **spp.**			
*Strep. thermophilus* YIT 2001 (ST-1)		Strong anti-oxidative activity against low-density lipoprotein (LDL) oxidation, high level of intracellular GSH, and anti-oxidative activity against LDL oxidation in hyperlipidaemic hamsters	[89]
* **Clostridium** * **spp.**			
*C. butyricum*		High intestine antioxidant enzyme (SOD, CAT, and GPx) activity and gene (hsp70 and ferritin) expression levels in shrimp fed with probiotics	[90]

**Table 6 biomolecules-12-01443-t006:** Principal methods used to evaluate probiotic antioxidant activities.

Methods	Reference
Oxygen radical absorbance capacity (ORAC assay)	[91,92]
Total antioxidant activity (TAA)	[93]
Reducing antioxidant power FRAP (ferric ion reducing antioxidant potential)	[94]
Lipid peroxidation inhibition assay TBARS assay or MDA assay β-carotene bleaching assay	[95]
Radical scavenging assay DPPH radical scavenging activity ABTS radical scavenging activity	[94,96]
Non-radical reactive oxygen species scavenging assay Hydrogen peroxide scavenging activity	[97]
Metal chelating capacity FRAP assay	[93]

**Table 7 biomolecules-12-01443-t007:** Some beneficial effects of antioxidant properties of synbiotics on human health.

	Synbiotics	Effects	References
Diabetes	*L. acidophilus + cinnamon powder*	Increase of antioxidant enzymes	[133]
	*L. acidophilus, L. casei,* and *B. bifidum* (6 × 10^9^ total CFU/g each) + 0.8 g/day of inulin	Increase of total antioxidant capacity and total GSH levels in diabetic patients under hemodialysis	[134]
Intestinal permeability	Multi-strain VSL3 # + FOS	Increase of catalase activity Protection of the mucosa from damage caused by chemical carcinogen and reduction of intestinal permeability	[33]
Ulcerative colitis	*B. infantis* + XOS	Inhibition of oxidative stress Downregulation of the proinflammatory cytokines TNF-α and IL-1β Upregulation of the anti-inflammatory cytokine IL-10 in the colitis-induced mice colon tissues	[131]
Immune systems	*L. lactis* SG-030 + GOS	Increase of the expression of tissue necrosis factor-α, interleukin (IL)-1β, IL-6, and iNOS synthase genes Increase the expression of P38, extracellular signal-regulated kinases, c-Jun N-terminal kinases, phosphoinositide 3-kinase, and Akt proteins	[135]
Hypercholesterolemia	*L. fermentum* MTCC + 5898-fermented buffalo milk	Reduced oxidative stress and inflammation in male rats fed with cholesterol-enriched diet	[136]

**Table 8 biomolecules-12-01443-t008:** Some antioxidant effects of synbiotics on animal health.

Subject	Synbiotics	Main Outcome	Reference
Poultry	*S. cerevisiae* + Mannanoligosaccharides (MOS)	Increased weight gain, reduced *E. coli* numbers in the small intestine and cecal digesta.	[171]
	Biomin^®^IMBOa	Improved body weight gain and feed conversion ratio, and protected against coccidiosis.	[172]
	*B. subtilis*, *B. licheniformis*, *C. butyricum* + yeast cell wall, + XOS	Increased average daily gain and breast yield, decreased feed/gain ratio and abdominal fat, and reduced MDA concentration in the thigh muscle, resulting in high-quality, oxidatively stable meat.	[143]
	*L. acidophilus, B. thermophilus, B. longum, Streptococcus faecium* + prebiotics	Increased serum overall total antioxidant capacity, and decreased serum total oxidant status and homocysteine concentrations.	[173]
	*B. subtilis* + XOS + MOS	Increased daily weight gain; feed efficiency; villus height; intestinal mucosa secretory IgA content; and antioxidant capabilities.	[145]
	*L. acidophilus* + garlic extract	Improved performance, intestinal health, antioxidants and nutrient digestion.	[174]
	*B. subtilis* + FOS	Improved average daily growth, FCR, reduced incidence of diarrhea and mortality.	[175]
Pigs	*L. plantarum*—Biocenol^TM^ LP96 (CCM 7512), *L. fermentum*—Biocenol^TM^ LF99 (CCM 7514) + flaxseed	Decreased lactate dehydrogenase leakage in the tissue extracts, and improved the immune status and the integrity of jejunum mucosa during infection.	[176]
	*Enterococcus faecium, L. salivarius, L. reuteri, Bifidobacterium thermophilum* + inulin	Decreased relative abundance of *Escherichia* in the ileum, cecum, and colon, and increased bifidobacterial numbers in the ileum.	[177]
	*L. plantarum* + maltodextrin and/or FOS	Reduced counts of *E. coli* O8:K88 in the jejunum and colon of piglets, and increased acetate concentrations in the ileum and colon.	[178]
	BiominR IMBO Pro/prebiotic, BIOMIN, GmbH Austria	Delayed the lipid oxidation process of the shoulder and ham samples during the refrigeration period.	[111]
Ruminants	*Ent. faecium* + lactulose	Decreased the ileal villus height, the depth of the crypts in the cecum, and the surface area of lymph follicles from Peyer’s patches.	[179]
	*Strep. faecium* + MOS	Improved fecal consistency and reduced the fecal score of calves without reducing in the number of scour episodes.	[180]
	Bioformula^®^	Improved average daily weight gain digestibility of dry matter and neutral detergent fiber and improved animal health.	[181]
	*S. cerevisiae* + Inulin	Increased pH in rumen, abomasum, and intestines, positively impacted the development of almost all morphological structures of rumen saccus dorsalis, rumen saccus ventralis, and intestine.	[182]
Aquacuture	*Ent. faecalis* + mannan oligosaccharides and polyhydroxybutyrate	Improved the growth performance and immune response of rainbow trout.	[183]
	*B. subtilis* WB60 + MOS	Improved growth performance, nonspecific immune responses, and disease resistance in Japanese eel.	[184]
	*Pediococcus acidilactici* + mannan oligosaccharides	Reduced MOS-induced gut humoral proinflammatory response by increasing the expression of some cellular-immune system-related genes, and reduced fish mortality after *V. anguillarum* infection.	[185]
	*Ped. Acidilactici* + pistachio hulls derived polysaccharide	Enhanced skin mucus and blood immune responses, upregulated immune-related genes expression, increased intestinal SCFAs content, as well as promoted antioxidative capacity.	[165]

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
