# Peer review of "Synbiotics and Their Antioxidant Properties, Mechanisms, and Benefits on Human and Animal Health: A Narrative Review"

_biomolecules, 2022, doi:10.3390/biom12101443_

Round 1

Reviewer 1 Report

The article is over all quite good. 

However, from first word I started a mental struggle because I’d never heard the word ‘synbiotic’ before.  It is so close to the word symbiotic. And, from their reference list, the earliest citation with the word ‘synbiotic’ was in 2015. It seems others may be confused about this word because an international association was convened in 2020 to define the scope of the word ‘synbiotic’.

Swanson, K.S., Gibson, G.R., Hutkins, R. et al. The International Scientific Association for Probiotics and Prebiotics (ISAPP) consensus statement on the definition and scope of synbiotics. Nat Rev Gastroenterol Hepatol 17, 687–701 (2020). https://doi.org/10.1038/s41575-020-0344-2

Although the authors cite this paper once, the way it is cited obscures any knowledge that synbiotic’ is a new word or that an international association had to be convened to define it lest there be misuse of the word. The authors use this citation glibly in support of their own very broad definition of the word ‘synbiotic’, by the following statement from line 77 of their paper. Therefore, please expand on the newness of the concept, and re-write this line so the boundaries of ‘synbiotic’ are stated.  It would also appear from the international association that the w

When probiotics are combined with prebiotics into formulations, the resulting mixtures constitute synbiotics [23].

The authors then go further to use the word as an adjective.  For instance, what is a synbiotic antioxidant in line 89 versus a antioxidant synbiotic’ in line 92?  And this begs a bigger question if the authors are redefining definitions previously given to the concepts of complementary synbiotics and synergistic symbiotics, given that the statement from the international association in 2020 about these terms says nothing about defining these concepts according to antioxidant capacities, as the authors vigorously do. Please rewrite and harmonize with the consensus statement from the international association.

 -------

Sentence 209 seems a poor start to the paragraph because the mechanisms are categoricaly delineated in the paragraph and multiple tables.  Certainly not ‘unknown’.

There are a lot of long sentences that need attention. 

Sometimes in the animal utility sections there are sentences that appear to relate to live animals, but the same is undermined when the sentences goes on to refrigeration storage of the meat.

Author Response

Thank you so much for all pertinent remarks and suggestions to our manuscript, which has been revised accordingly, and all corrections have been highlighted in the revised version. 

Please expand on the newness of the synbiotic concept, and rewrite the statement from line 77 so the boundaries of 'synbiotic' are stated.

The synbiotic concept from line 77 has been expanded as follows:

"When probiotics are combined with prebiotics into formulations, the resulting functional products constitute synbiotics. Even though the synbiotic concept was first described 25 years ago, the panel of International Scientific Association for Probiotics and Prebiotics (ISAPP) recently updated the synbiotic definition as “a mixture comprising live microorganisms and substrate(s) selectively utilized by host microorganisms that confers a health benefit on the host” [24]. Such a preparation can be designed in complementary to target the host microorganisms, or in synergism for which the prebiotic is selectively utilized by the co-administrated probiotics to achieve one or more health benefits. The term synbiotic is often confused with symbiotic, which refer to an ecological relationship in a natural ecosystem with two organisms (the symbiont & the host) in symbiosis."

What is 'a synbiotic antioxidant' in line 89 vs. a 'antioxidant synbiotic' in line 92?
Please rewrite and harmonize with the consensus statement from the ISAAP.

Sentences from Line 102 to Line 108 have been rewritten and harmonized as follows:

"In fact, it is important in the case of synbiotics with antioxidant properties to distinguish those from prebiotics, probiotics and their metabolites, or those from bioconverted prebiotic compounds. Two main types and mechanisms may be involved : (i) complementary synbiotics for which prebiotics and probiotics act independently with additive effect as antioxidants at the host [27]; (ii) synergistic synbiotics where prebiotics are antioxidants or not, while supporting and enhancing the probiotics antioxidant performance for generating higher properties than each component "

Sentence line 209:

    The introduction sentence in line 209 is not necessary, and has been removed.

There are a lot of long sentences that need attention

    Long sentences have been divided in shorter ones.

Sometimes in the animal utility sections there are sentences that appear to relate to live animals, but the same is undermined when the sentences go on to refrigeration storage of the meat

    The sentence has been corrected for avoiding confusion (Line 518-522) as follows:

"Furthermore, it was discovered that a synbiotic-supplemented diet reduced MDA levels in broilers. The addition of bee pollen and propolis extracts in feed mixtures, in combination with probiotics added into drinking water for broiler chickens, also reduced oxidative processes in the breast and thigh muscles during 7-days of chilling storage [147,148]."

Reviewer 2 Report

his paper reviews synbiotics and their antioxidant properties, mechanisms and benefits on human and animal health. Recent years, probiotics, prebiotics and synbiotics have been a hot research area, and their antioxidant activity and many other physiological activities have been widely reported. Therefore, the review of synbiotics and their antioxidant properties is a very meaningful work. This manuscript is well organized and written. I will just make a few comments below.

- Abstract: This is more like an introduction or background than an abstract. The abstract should contain all the important information of the article, such as the research background, main review results, final conclusions and prospects.

- Line 70-72: Please mark the references cited here.

- Table1 should be before Fig 1.

- Fig 1 is too simple to elucidate the concepts and functions of prebiotics, probiotics and synbiotics, which is very significant in the article.

- Table 4: Please mark the references.

- Table 7: “B. bifidum (2 × 109 CFU/g each)” may has a spelling mistake.

Author Response

Thank you so much for all pertinent remarks and suggestions to our manuscript, which has been revised accordingly, and all corrections have been highlighted in the revised version.

Abstract

The abstract has been rewritten.

"Antioxidants are often associated with a variety of anti-aging compounds that can ensure human and animal healthy longevity. Foods and diet supplements from animals and plants are the common exogenous sources of antioxidants. However, microbial-based products, including probiotics and their derivatives, have been recognized for their antioxidant properties through numerous studies and clinical trials. While the number of publications on probiotic antioxidant capacities and action mechanisms is expanding, that of synbiotics combining probiotics with prebiotics is still emerging. Here, the antioxidant metabolites and properties of synbiotics, their modes of action, and their different effects on human and animal health are reviewed and discussed. Synbiotics can generate almost unlimited possibilities of antioxidant compounds, which might be with superior performance than those of their components through additive or complementary effects, and especially by synergistic actions. Either combined with antioxidant prebiotics or not, probiotics can convert these substrates to generate antioxidant compounds with superior activities. Such synbiotic-based new routes for supplying natural antioxidants appear relevant and promising in human and animal health prevention and treatment. A better understanding of various component interactions within synbiotics is a key to generate higher quality, quantity, and bioavailability of antioxidants from these biotic sources".

Line 70-72

The reference has been cited in the right place.

Table1 should be before Fig 1

Fig1 has been placed before Table 1

Figure 1

Fig 1 has been completed for elucidating the concept of pro-, pre-, and synbiotics in relation to their antioxidant properties. 

References

References have been added in the legend of Table 4.

Table 7

The term "2x10^9 CFU/g each" has been replaced by "6 x10^9 total CFU/g" for the three strains.